# Needs Assessment for a VR-Based Adult Nursing Simulation Training Program for Korean Nursing Students: A Qualitative Study Using Focus Group Interviews

**DOI:** 10.3390/ijerph17238880

**Published:** 2020-11-29

**Authors:** Jaehee Jeon, Jin Hee Kim, Eun Hee Choi

**Affiliations:** 1Department of Nursing, Gangneung-Wonju National University, Gangwon-do 26403, Korea; 2Department of Nursing, Doowon Technical University, Gyeonggi-do 17520, Korea; friziajini@doowon.ac.kr; 3Department of Nursing, Korean Bible University, Seoul 139-791, Korea; ichoi9201@naver.com

**Keywords:** student, nursing, virtual reality, focus groups, interview

## Abstract

Virtual reality programs are being actively utilized in various education fields, but not many have been developed/used in nursing. This study aimed to explore the essential components and improvements needed in an adult nursing VR-based simulation training program for nursing students through focus group interviews (FGIs). This was a qualitative study. Fourteen nursing students from three cities in Korea who had experienced clinical practice and simulation training participated. They were divided into three FGIs. Data were collected from February–March 2020. We analyzed the data from the FGIs using Colaizzi’s phenomenological methodology. In total, 40 themes emerged, divided into 13 theme clusters and the following four categories. When developing an adult nursing VR-based simulation training program, the development should focus on addressing the limitations of conventional clinical practice, and these should be analyzed; it should also reflect students’ needs, including the following: provide an array of scenarios/skills to be trained; difficulty-specific learning scenarios; immediate feedback (e.g., those in computerized games); simulate emergency situations; simulate clinical cases that are difficult to experience in clinical practice; and allow for the training of patient–nurse communication skills.

## 1. Introduction

Virtual reality (VR) is a cutting-edge technology that refers to a specific environment that resembles reality, but is not real; instead, it integrates a variety of engineering information technologies that allow for a computer to create a virtual space [1]. VR programs, in turn, are interactive simulations that are designed for users to experience situations that resemble real-world experiences, but through the use of computer hardware and software; accordingly, these programs can be applied to a number of fields [2]. For example, various fields of research have been exploring VR-related programs, including studies in the context of tangible, entertainment, education, and industrial content, as well as virtual experience marketing for products. [1]. In the field of medicine, VR-based contents and technologies are usually applied to cultivate competent specialists by training medical students, interns, and residents not only in conducting less invasive procedures (e.g., coronary angiography and endoscopy), but also in conducting surgeries (e.g., appendectomy and laparoscopic) [2,3,4,5]. Corroborating augmented reality activities utilizing VR content have already been found to significantly improve students’ learning [6,7].

Recently, population aging has led to an increase in the number of patients with chronic diseases (e.g., cancer and cardiovascular and cerebrovascular diseases), and this increase has escalated the need for healthcare providers equipped with highly advanced skills and knowledge [3]. Thus, skilled physicians and nurses are needed in clinical practice. Particularly, a study remarked that proficient nurses play a critical role in monitoring patients’ symptoms 24 h a day, detecting abnormalities, assessing problems, and promptly responding to various situations involving the patient [8]. Actually, the appropriate allocation of skilled nursing in clinical settings has been reported to significantly reduce patient hospitalization, mortality, and complication rates [9,10]. Therefore, currently and internationally, the demand for skilled nurses is consistently high in clinical settings.

Accordingly, to allow for Korean students to become skilled nurses after graduation and be able to effectively transition to clinical settings, nursing schools in Korea are designing their curricula such that students learn advanced theoretical knowledge through various courses over four years and complete more than 1000 h of clinical practice [11]. However, with the increase in patient rights in clinical practice and to prevent the violation of these rights, Korean nursing students are currently permitted to provide only a limited number of actual medical practices during clinical practice; this often limits their clinical practice experiences to the simple observation of nursing activities as they are delivered/provided by other healthcare professionals [11,12]. Additionally, although more than 230 nursing schools have been established in Korea, only 10% have an affiliated university hospital; this means that the clinical practice hospitals for the other 90% are likely to have poor environments, denoting that the quality of clinical practice experiences given to nursing students substantially varies according to the presence or absence of an affiliated university hospital [13,14]. Therefore, even after completing four years of curriculum, it is usual for newly graduated nurses to be able to start working only after completing training again for a certain period of time [12,15]. Consequentially, existing skilled nurses have an increased workload, and newly graduated nurses are exposed to high work stress and burnout, ultimately leading to high turnover; this situation has been identified as a problem in various clinical settings [15].

To solve this problem, for the past 10 years, nursing schools have been incorporating simulation training in their nursing curriculum [13,16]. Simulation training using a high-fidelity simulator enables students to experience an environment that is very similar to the actual clinical setting and to provide nursing interventions in different scenarios, such as evaluating patients’ states, discovering health problems, and providing appropriate care. During this process, students can get involved in a realistic experience—in contrast to theoretical education—thus enabling them to have a better grasp as to how they should assess patients’ illnesses and symptoms, and how to interact with patients [16]. However, simulation training has several limitations; first, the simulator has little vitality, so it cannot resemble an actual person. Furthermore, there are many limitations regarding the healthcare and nursing care situations that can be presented in a simulator, and regarding the interactions with and responses from a patient created by it [17,18]. VR-based simulation training is a practice by an interactive program that allows students to experience nursing interventions autonomously and repeatedly, regardless of time and place. The virtual environment is set in the medical field with various patient situations, without the need for external facility settings.

Despite these limitations, studies have shown that VR-based simulation training programs have been utilized to cultivate professionals in diverse fields, including physicians, teachers, and soldiers [2,3,4,5]; still, VR-based simulation contents and programs for nurses have been lacking. To develop such programs for nurses, it seems necessary to assess the problems in conventional simulation training and identify the needs of nursing students regarding these programs—namely, the needs of the user. Therefore, this study aimed to explore the essential components and improvements needed in an adult nursing VR-based simulation training program for nursing students through focus group interviews (FGIs).

## 2. Materials and Methods

### 2.1. Study Design

This study was a qualitative investigation using FGIs.

### 2.2. Sample

The inclusion criteria were the following: being a nursing student; providing voluntary informed consent; having experience in simulation training in adult nursing; and having completed at least two semesters of clinical practice. We made recruitment announcements in three nursing departments in three provinces in Korea. Although sample sizes are usually not pre-determined for qualitative studies, most studies with this design collect data until saturation. In the present study, first, we recruited 14 participants; based on Morgan [19], we then divided the sample into three groups of 4–5 participants according to the level of structuring of the study topic, the appropriate number of focus groups for saturation, and to ensure low heterogeneity of participants in each group. After completing the interviews for the three groups, we did not recruit additional participants; this decision was made based upon judging that the data had reached saturation.

### 2.3. Procedure

After obtaining approval from the International Review Board (IRB), we arbitrarily selected three universities in three cities in Korea, and requested consent and cooperation from the heads of the respective nursing departments. Then, we posted an announcement on the bulletin board about the purpose and procedure of the study. After recruitment, we collected the data from the volunteers.

#### 2.3.1. Development of Interview Questions

First, we examined the study aims. Next, to study the topic as effectively as possible by exploring the basic background information and using key questions, we conducted interviews based on Breen’s guidelines [20]. Breen’s guidelines provide practical guides for each step of FGIs. It explained step by step how to proceed with FGQ’s interview questions, which this study followed to construct interview questions. By employing a literature review and discussions prior to the interviews, the research team developed draft questions. The key interview questions are listed in Table 1.

#### 2.3.2. Interview Procedure

Data were collected from February to March 2020. The interviews were conducted using the following procedure:Preparation: The FGIs were conducted in a quiet, isolated lecture room in the respective school. Thirty minutes before students’ arrival, the facilitator, assistant, and field note-taker came to the interview room to arrange the seats and prepare for recording. The researchers prepared snacks and allocated them across the room to help relax the mood and allow for participants to enjoy the snacks before the interviews. Upon students’ arrival, prior to anything else, the researchers handed out sheets that contained information about the study; then, after providing a detailed oral explanation of the study purpose, method, and data management and disposal, the researchers asked for participants to sign the written consent form, which was obtained for each participant. Prior to discussion onset, the researchers surveyed participants’ general characteristics.Introduction: One researcher served as the facilitator for the interviews. First, the facilitator explained to participants that all information obtained during the discussions would only be used for research purposes; then, after obtaining consent for recording, the portable recorder was turned on.Discussion: The facilitator put forward topics according to the developed interview questions and fostered a climate that allowed for participants to freely discuss the topic. The interviews went on until reaching data saturation (i.e., no new contents or statements emerged), which took about 1.5–2 h. Throughout the interview, the facilitator made sure that each participant was provided with sufficient time and equal opportunity to speak; the facilitator also made use of nonverbal communication (e.g., nodding and smiling) to encourage all participants to actively partake in the interview.Wrap-up: The interview contents were reviewed based on participants’ responses, and the participants were given an opportunity to add any comments. Furthermore, after the interview’s conclusion, the researchers thanked the participants and informed them that another interview round would be conducted if necessary. The participants were given a small gift (gift card).

### 2.4. Ethical Considerations

This study was approved by the IRB at the researcher’s affiliated university (No. GWNUIRB-2020-4). Given that study participants were nursing students, our sample was recruited from schools that had no affiliation to any of the researchers; namely, we submitted requests for participant recruitment at an unaffiliated institution. Prior to the FGIs, the principal investigator informed participants about the study purpose, contents, recording, and the transcription of interviews; then, the investigator explained that participants could withdraw from the study at any time, without any setbacks, and that the study data would be retained in a locked cabinet for three years following the completion of the study before disposal. The interviews were conducted only after obtaining written consent from participants. All participants who completed the interviews were given a small gift voucher. While transcribing the interviews, we coded participants’ personal information anonymously.

### 2.5. Data Analysis

The analysis of the contents of the participants’ statements and field notes follows the analysis procedure of Colaizzi’s phenomenological methodology [20,21]. It is designed to extract meaningful sentences or phrases from the contents described by participants, form general and abstract statements based on them, construct meanings, categorize them into thematic collections, and describe the essential structure of the experience. First, one research assistant transcribed the recorded interviews; then, the principal investigator reviewed the accuracy of the transcripts.

Second, to gain an accurate understanding of the interview contents, the transcripts were repeatedly read, and meaningful statements (i.e., with sentences as the unit of analysis) for each key question were underlined.

Third, the principal investigator categorized the key contents in the meaningful statements/expressions and grouped them into larger categories. Then, the investigator removed duplicate expressions, and the statements were re-stated in general and abstract form.

Fourth, the investigator structured meanings from meaningful statements; from these structured meanings, the researchers formed themes, theme clusters, and categories. Finally, the common components of the experiences were integrated to define the fundamental thematic structure.

### 2.6. Preparation of Researchers

The researchers have 13–24 years of clinical experience in a tertiary hospital and are the current faculty of a nursing school in Korea. We all have conducted a qualitative research course in graduate school, attended several conferences and seminars on qualitative research, and have conducted qualitative studies in the past.

### 2.7. Rigor of the Study

To increase the qualitative rigor of the study, this study was conducted in accordance with the four criteria proposed by Lincoln and Guba [22]: credibility, fittingness, auditability, and confirmability. To ensure credibility and that there were no omissions, all researchers listened to the recordings several times and compared them with the transcripts; if there was any inaccuracy in the transcript, the researcher went on to verify the content with the relevant participant to ensure data credibility.

To ensure fittingness, we selected participants appropriate for the study purpose and conducted FGIs until reaching data saturation.

To ensure auditability, all researchers comprehensively reviewed the transcripts and analyses to establish the confirmability of the findings, and a review of the study was requested from a second qualitative study expert. Furthermore, we recorded the interview data collection procedure in detail, as described above.

To ensure confirmability, we recorded the entire data collection procedure and retained interview recordings and transcripts to allow for these data to be reviewed by any researcher at any time.

## 3. Results

### 3.1. Participants’ Characteristics

In total, 14 nursing students (i.e., eight females and six males) who were aged 22–27 years and had experienced adult nursing simulation training participated in this study (Table 2).

### 3.2. Content Analysis Results

To grasp the overall meaning of the interviews, all researchers repeatedly read the transcripts. Participants gave statements about various aspects of their experiences of clinical practice and simulation training, as well as about VR-based simulation training. Hence, to extract the unit of analysis, we established the three following criteria: clinical practice, simulation training, and VR-based simulation training.

According to the criteria, we extracted 154 units of analysis, which were then open-coded into three larger codes (i.e., clinical practice, simulation training, and VR-based simulation training) and seven smaller codes (i.e., difficulties with clinical practice, pros and cons of simulation training, direction of improvement for simulation training, expectations for VR-based simulation training, need for VR-based simulation training, and direction of development of VR-based simulation training). Then, we categorized the coded units of analysis into more abstract categories; accordingly, the need for VR-based simulation training program was categorized into 40 themes, grouped into 13 theme clusters, and further categorized into the four following categories: limitations of clinical practice, benefits of simulation training, need to improve simulation training, and need for VR-based simulation training (Table 3).

#### 3.2.1. Category 1: Limitations of Clinical Practice

This category comprised the three following theme clusters: “Feeling awkward”, “Confounded”, and “Limited training.” Participants reported that the first clinical practice was especially more difficult, that they felt awkward around patients and nurses, and that they were embarrassed and baffled when faced with unexpected situations. Moreover, they were not satisfied with the limited training they were given, during which they were often only allowed to take vital signs or simply observe.

1.Feeling awkward. This theme cluster comprised the two following themes: “Feeling awkward for getting in the way of nurses’ work” and “Patients do not want students’ participation in care.” Participants felt as if their presence would hinder care delivery to patients by nurses, and experienced situations in which patients did not want student participation, mostly owing to privacy reasons. Particularly, male students were sometimes asked to stand outside of the room because caregivers or patients felt uncomfortable around them. Information within square brackets were added by the researcher to ensure the complete understanding of the sentences. Here are some excerpts that represent these themes:

“When I was not yet used to clinical practice, I was always [walking] on eggshells because I felt that, when I observed or stayed next to the nurses when they gave nursing care, I could get in their way…” (Participant F)

“Male nursing students could not stay around to observe female training, even when the instructor said it was okay, because the caregiver or patient felt uncomfortable; and female nursing students had to stay behind the curtain when there were many male patients and caregivers; and patients refused [student participation] even if the instructors said it was okay.” (Participant D)

2.Confounded. This theme cluster comprised the two following themes: “Flustered by an unexpected situation or care instruction” and “Difficult to adapt to training because many parts were not covered at school.” Participants reported that they felt flustered by situations that they had not learned about in school and in which they had to give unexpected instructions. Particularly, they had a hard time adapting during their first clinical practices. Here are some excerpts that represent these themes:

“When I was training at the hospital, there were situations that I had not expected, and when the instructor said something [about the unexpected situation], I would not be able to understand because I had not learned about it in school, so I could not answer; so I was really embarrassed so many times.” (Participant I)

“During clinical practice, I really did not know what to do to help or the order of work, so I just tried to stay right behind the instructor all the time to try and help in some way, like getting things she needed; and that’s about it.” (Participant H)

3.Limited training. This theme cluster comprised the two following themes: “Limited training that only addresses some skills” and “Only able to observe in most situations.” Participants felt that the training was extremely limited, as they were often only allowed to take vital signs or a blood sugar test during training; thus, often, all they could do was to follow the nurses around. Here are some excerpts that represent these themes:

“Some of the limitations were that, since we are nursing students, there were not many things that we could do ourselves; and, even when they let students do things, the only things we could do were taking blood sugar test and blood pressure. So, since things were the same at every training, we get better at those skills, but the remaining core skills we could only practice in school.” (Participant K)

“Many emergencies occur in the hospital, but even in those situations, we could not readily volunteer to do something, so we observed what the nurses did; but, I think there is a big difference between just watching and trying it yourself.” (Participant B)

#### 3.2.2. Category 2: Benefits of Simulation Training

This category comprised the three following theme clusters: “Extremely helpful because it can be applied to clinical practice”, “Become confident in clinical practice”, and “Improve on undeveloped nursing skills.” Participants reported that, as they overcame the limitations of clinical practice and learned what to do during clinically rare cases and emergencies in a simulated environment (which closely resembled the actual clinical setting), they experienced an integration between the theories they had learned and real-world situations.

1.Extremely helpful because it can be applied to clinical practice. This theme cluster comprised the three following themes: “Learning what is appropriate to do in clinical settings with hands-on practice”, “Things learned in simulation training are helpful in clinical practice”, and “Learning in advance helped gain a different perspective during clinical practice.” Participants felt that, by using simulation training, they were being more appropriately trained to deliver care in the actual clinical setting, mainly because they were able to practice core skills for each situation and make their own judgments. Additionally, through simulation training, participants learned what they needed to focus on during clinical practice and developed goals related to nursing care delivery. Specifically, they reported that their spectrum of knowledge broadened because they were given the opportunity to observe patient symptoms that they had not thought of before, something that made them become more interested in the results of the tests. Here are some excerpts that represent these themes:

“After observing it [a clinical situation] in a clinical setting and doing it [delivering nursing care] hands-on in simulation training, [it] helped [me to] integrate the theoretical aspect and practical aspect; so, the learning effects were maximized.” (Participant F)

“After gaining experience in the simulation training, I developed a type of virtual blueprint in my head about what I need to do; like, what to prepare for [when delivering nursing care] and the order of things that I need to do for a patient; so, I reviewed the things [to which] I need to prepare for when I go to clinical practice, so that I can deliver prompt assistance and nursing care in the order that I know; so, I think I liked that simulation training can be applied to clinical practice.” (Participant L)

“Because I had done the simulation training and was taught by the professor before I go to clinical practice, I was more prepared and was able to see more things; and, I think I was able to understand things like [the reason] why we were running a particular test on a patient in a [specific] situation.” (Participant I)

2.Become confident in clinical practice. This theme cluster comprised the two following themes: “Able to give patients an explanation after simulation training” and “Dealing with patients became easier after simulation training.” The participants were able to explain things they had experienced during simulation training with confidence. In particular, they experienced that their spectrum of knowledge broadened, as they observed patient symptoms that they had not thought of before and became more interested in test results. Here are some excerpts that represent these themes:

“Because I underwent simulation training before clinical practice, for example, I was able to instruct patients on my own with terms, like on how the patient should breathe and how this [care delivery] should be done, and I think these things are the ones that have changed after performing simulation training before clinical practice.” (Participant G)

“You know, it was easier for me to understand the patient’s history because I practiced it in simulation training before I went to clinical practice, so I think simulation training is essential, should it be in adult, psychiatric, or pediatric nursing.” (Participant D)

3.Improve on undeveloped nursing skills. This theme cluster comprised the two following themes: “Undeveloped nursing skills are discovered by hands-on practice” and “Review mistakes through debriefing.” Participants stated that they discovered their undeveloped nursing skills as they delivered direct patient care and learned what skills they needed to improve through feedback during debriefing. Here are some excerpts that represent these themes:

“I think with hands-on practice, I was able to look back at myself; like, [at] my weaknesses and strengths…” (Participant F)

“What I especially liked about simulation training was that we watched the training video together during the debriefing session, and other students gave me feedback about what I missed and what I did wrong; and that I really liked.” (Participant M)

#### 3.2.3. Category 3: Need to Improve Simulation Training

This category comprised the four following theme clusters: “Not so realistic”, “Difference in effectiveness according to the instructor”, “Want to experience diverse clinical situations”, and “Want more simulation training opportunities.” Participants stated that simulation training had the two following major problems: it was unrealistic and it forced students to engage in pre-established behaviors. Most of all, they were dissatisfied with the fact that instructors’ widely varying teaching competences affected the effectiveness of their training, and with the fact that, although they wanted to experience a wide variety of cases, they were not provided with such enriched content.

1.Not so realistic. This theme cluster comprised the three following themes: “Carelessly treat the unrealistic subjects in the simulation”, “Subjects in the simulation cannot give feedback”, and “It is regrettable to be confined to only doing the actions determined according to the scenario.” Participants mentioned that one shortcoming of simulation training is that the role of the patient is played by a virtual model, which means that they cannot have a real conversation with it, thereby making them feel as if they were not delivering actual care. Most of all, they felt that not being able to receive feedback from the patient was a serious disadvantage. Here are some excerpts that represent these themes:

“We are not giving care to an actual human, but to a model, so we are less tense and we treat it more carelessly, [either] knowingly or unknowingly.” (Participant J)

“You know this is a model, and the professor just gives us a voiceover and lays out the situation, so you know the model will just remain still. Therefore, I think one downfall was that it was not so realistic.” (Participant A)

“What I did not like about the simulation training was that I could not get direct feedback from the model patient, so I could not know what the patient thought and stuff [like that].” (Participant N)

2.Difference in effectiveness according to instructor. This theme cluster comprised the two following themes: “Teaching method differs across instructors” and “Competence varies across instructors.” Participants stated that they wanted the instructors to use a standardized method of instruction for simulation training and that they had learned different things with varying levels of effectiveness according to the instructor. Here are some excerpts that represent these themes:

“[The form of] Acting and the contents of teaching differ across the professors, so I think it was difficult to know the flow by which we should have gone along and how we should have behaved and stuff, and it would be great if those kind of things could be standardized.” (Participant G)

“The professors have to act, so some are kind of awkward in their acting, while others are good, and depending on that, we kind of learn different things with different effectiveness, and this is a shortcoming.” (Participant J)

3.Want to experience diverse clinical situations. This theme cluster comprised the three following themes: “Want to go to clinical practice only after learning how to deal with urgent situations”, “Experiencing rare diseases helps during clinical practice”, and “Want to learn how to talk to patients and caregivers.” Participants mentioned that learning how to deal with emergencies and with situations that they cannot often experience in clinical settings in advance would be helpful for their training. Particularly, they wanted to experience cases that are relatively rare and learn how to converse with patients. Here are some excerpts that represent these themes:

“If you are faced with emergency situations, like if the patient starts acting out or if the patient’s state suddenly worsens, in advance, would not you be more flexible in dealing with them in clinical settings later?” (Participant D)

“I think it would be better if you could also learn about communication, like how you can comfort the patient by engaging in conversation while doing your role as a nurse and how to interact with caregivers.” (Participant H)

4.Want more simulation training opportunities. This theme cluster comprised the two following themes: “Not enough opportunities to undergo simulation training” and “Want to be able to choose the desired case.” Participants were frustrated with the fact that, although they wanted to choose various cases/courses, they were either never offered the opportunity for such choice or were given the opportunity to choose only among a small sample. Here are some excerpts that represent these themes:

“There was only one semester offered [for simulation training], so I was sad that I could only try this once.” (Participant B)

“I think it would have been better if I was able to experience several other courses as well.” (Participant E)

“In my school, you are only allowed to choose one course, so students who chose adult nursing were able to experience emergency situations, like CPR, while those who did not experience them before went to clinicals; so, not being able to practice how to deal with those emergencies was a downfall.” (Participant C)

#### 3.2.4. Category 4: Need for VR-Based Simulation Training

This category comprised the three following theme clusters: “Great expectations for VR-based simulation training”, “Using desired scenarios during VR-based simulation training”, and “Various ways of running VR-based simulation training.” Participants expected that VR-based simulation training would be more realistic and fun because it would provide a situation that more accurately resembles the clinical setting, and they also expected that this type of training would be standardized. Once more, they remarked about the desire to experience cases which they could not often experience in clinical settings, to learn about how to deal with emergencies, and to improve the learning effects of the training by being able to use specific program compositions based on student level. Regarding the timing of training, students mentioned that it should be applied before, after, or both before and after clinical practice, and they wished to choose from and learn about cases that interested them.

1.Great expectations for VR-based simulation training. This theme cluster comprised the five following themes: “It will be helpful in learning skills”, “Everything in the simulation feels real”, “Diverse scenarios can be used, so it will be fun”, “Relieve the gap in knowledge between theory and practice”, and “Standardized simulation training.” Participants wished that the simulation felt real because this would allow for them to learn specific skill sets that are needed for specific situations and feel like they were actually treating patients. Furthermore, they thought that this type of training would be fun because it would provide diverse situations, and while there are limitations in using supplies (e.g., fluid set, intravenous catheter, syringe, etc.), during training—unlike in real clinical settings—they expected this problem to be resolved with VR-based simulation training. Most of all, they were hopeful that this type of simulation training could be standardized if a universal program for all schools was developed. Here are some excerpts that represent these themes:

“If we use VR, I think we would really be able to make adjustments when giving fluids to patients, and, you know, things like setting the fluid gtt [drops] and stuff are really important. So, would not we be able to focus on skills like that and do them correctly?” (Participant B)

“In regular simulation training, the model cannot realistically express higher levels of pain, such as sweating and shivering, and stuff, and we cannot observe these [signals]. Therefore, in VR, if the machine starts beeping or the patient evidently starts shivering or turns blueish, and if this emergency situation is expressed more realistically, would not we be able to better understand that, ‘oh, this is really an emergency?’” (Participant C)

“It would be more fun for students for sure, because, right now, we just talk to this patient model that is lying on the bed and say things like, ‘Hello, how are you?’ and this is boring; but, if we use VR, it would be more realistic and we would be given more diverse situations, which would be more fun for students.” (Participant K)

“If we use VR, we could potentially be placed in an environment that we have no knowledge of, so [that] we [could] feel like we were in unfamiliar hospitals, or wards, or special units. And, I think [that] being able to change the background and environment is kind of attractive.” (Participant J)

“It would differ across schools, but, you know, you have some restrictions in the supplies that you can use. If we use VR, I think it would be great if the simulation could more realistically show the supplies that are widely used in actual hospitals but that often cannot be seen or used in school.” (Participant J)

“The number of simulation training sessions you go through differs across schools, and the method is not standardized. If VR is introduced, then I hope that all schools and all nursing schools can use it, and that students can be more effectively trained by standardized simulation environments.” (Participant M)

Using desired scenarios during VR-based simulation training. This theme cluster comprised the eight following themes: “Partake in an environment just like the clinical setting”, “Phased learning scenarios”, “Immediate feedback, like in games”, “Cases difficult to experience in clinical settings”, “How to deal with urgent situations”, “Practice core skills in real-world–like settings”, “Practice communication”, and “Learn how to operate machines.” Participants wanted the VR-based simulation to provide an environment that closely resembled the clinical setting, and wanted to play the role of the nurse that provides direct care to the patient. Additionally, they wanted the simulation to feature varying levels of difficulty and to allow them to go on to the next step only when the correct treatment was given, preferably by giving immediate feedback, such as what happens in computerized games. Particularly, they wanted to experience a variety of cases that they would not often encounter in clinical settings and learn how to deal with emergencies. They expected that their core skills would be improved by VR-based simulation training because they would perceive the training as if they were in a real situation in which they needed to employ core nursing skills on a patient. Moreover, they were highly interested in this type of training, and stated that they wanted to learn the basic skills that newly graduated nurses should be equipped with (e.g., communication and handling of nursing-related machines). Here are some excerpts that represent these themes:

“I think it would be good if the simulation can provide an environment where there is a ward with 4–5 patients and you can [deliver] care for these patients as the nurse in charge of them.” (Participant J)

“If VR is implemented, I think [that] experiencing patients that we have not encountered in this training room would be a really valuable experience, so I really wish that VR gets implemented.” (Participant H)

“The situations where we were really flustered were [those] when the patient suddenly had a drop in SpO_2_ and the nurses raised the patient’s upper body and suctioned, but we only tried suction when the patient had no phlegm; so, I think it would be really good to try suction in patients that actually have the crackling sound of phlegm.” (Participant B)

“I think VR is better in terms of skills because, with a model, you can stick the intravenous needle a few times without a problem; but, with VR, if the patient shows a response like a blood pressure drop when you insert the needle incorrectly, this can make you become more tense and focused during the training, so your skills would improve.” (Participant D)

“If VR is implemented, then I think it is important [that the simulation enables us] to talk to patients as much as to judge the situation and deliver care; so, I think it would be better if we could talk to the patients because it is a simulation training.” (Participant N)

2.Various ways of running VR-based simulation training. This theme cluster consisted of four themes: “Want to learn how to deal with important situations in clinical practice prior to the actual training”, “Want to undergo VR-based simulation training after experiencing the actual clinical setting”, “Need to undergo simulation both before and after training”, and “Run simulation as an elective for advanced learning.” Many participants stated that, although VR-based simulation is important before clinical practice, undergoing it after experiencing the clinical setting would be more effective. Some also stated that VR-based simulation should be applied both before and after clinical practice. Others wanted to experience cases that they would not easily experience during clinical practice through VR-based simulation, and they also wanted to be able to choose the cases which they would encounter. Here are some excerpts that represent these themes:

“The greatest benefit of VR is the visual aspect, and if you experience it before going to clinicals, and then see how things are done in a clinical setting, then you can compare VR with clinical practice.” (Participant F)

“I think the standard should be the actual clinical setting; so, it would be better if we experience clinical setting first and then undergo VR simulation.” (Participant H)

“VR is not a real-life situation and its purpose is to practice, so I think our views will broaden if we try it before clinicals and if we do it after clinicals; then, we would think, ‘oh, this kind of situation can also happen,’ ‘I should see if I can do this,’ or ‘I should see if this actually happens in the clinical setting’; so, I think the timing is not really important.” (Participant C)

“I think that, with VR, there would be more limitations in things you can do hands-on compared with the currently utilized simulation training; so, it should be run as an advanced learning course to complement the shortcomings of existing simulation; [it should be used] as an elective course, not a required course.” (Participant F)

## 4. Discussion

This was a phenomenological study that assessed the needs regarding VR-based simulation training programs for adult nursing of Korean nursing students using FGIs. Our study participants reportedly experienced various limitations during their clinical practice, stating also that the simulation training helped them gain confidence in clinical practice, mainly because it maximized the effects of training and was translatable to clinical practice. However, they remarked that the simulation training had the following limitations: it lacked realistic portrayals of patients and environments; provoked pre-established behaviors based on the scenario; had varying effectiveness depending on the instructor; and provided limited clinical situations. Thus, participants had varied expectations regarding the implementation of VR-based simulation training, proposing ideas that this type of training should, optimally, be realistic, practical, provide diverse clinical scenarios, and include different methods to run the simulation program.

Regarding the theme cluster limitations of clinical practice, and its themes of feeling awkward, being confounded, and perceiving the limitations of clinical practice, previous qualitative studies have reported similar trends; namely, that nursing students were dissatisfied with clinical practice because they were mostly running small errands and had to remain passive observers during training [12,23]. Moreover, according to a study that explored the clinical practice experiences of newly graduated nurses during their school years, although students showed increased pride as they underwent various experiences through clinical practice, they reportedly felt proud only after applying their theoretical knowledge hands-on or experiencing it directly [12]; thus, the literature and our results underpin the need to provide more opportunities for nursing students to practice various skills and situations during clinical practice, instead of being simple observers of other nurses’ clinical practice.

Recently, various changes in the clinical environment and the rapidly evolving healthcare environment—which directly highlight the limitations of observation-centered clinical practice and its narrow learning outcomes—have made nursing schools in Korea start to enthusiastically utilize scenario-based simulation trainings to teach nursing students the knowledge, skills, and attitudes essential for clinical competence; accordingly, various standardizations have been developed and presented for simulation nursing education, thereby evoking its gradual systematization, which may in turn allow for this type of education to promote positive learning outcomes and the translation from theory to practice [24]. In the present study, nursing students mentioned that they had mostly positive experiences with simulation training; reportedly, they were able to apply this type of training to clinical practice, which in turn allowed them to maximize the effectiveness of training, gain confidence in clinical practice, and foster their less developed nursing skills. One study has reported that simulation training is effective in most subjects in the nursing field, including pediatric, adult, psychiatric, and maternal nursing [25]. Particularly, it has been found to be effective in improving nursing students’ critical thinking and problem-solving skills, mostly by providing opportunities for them to experience diverse roles in emergencies and situations involving seriously ill patients [14,17,18]. However, despite the benefits of simulation training, the participants also expressed the need to experience more diverse and realistic clinical situations in these virtual training environments; two of the major complaints about the traditional simulation training related to it being unrealistic and a facilitator for the repetition of limited, pre-established behaviors. Furthermore, and concurring with prior research [26], students in our sample stated that the effectiveness of the training differed by instructors’ teaching experience and students’ active participation. Thus, it is necessary to implement features in simulation trainings that are more compelling for students, namely, that promote their active engagement; this remark is similar to that presented in a previous study [27].

For this very reason, efforts have been made to advance nursing simulation training by utilizing high-fidelity and VR-based simulations [27,28]. However, existing nursing VR-based simulation training is limited to online programs, such as the following: the vSIM^®^ program, which runs scenarios in a web-based platform [29,30], and a program that runs a virtual scenario using an online lecturing module [31]. Nonetheless, even these programs have rarely been utilized in the field of adult nursing [29,30]. Contrarily, a variety of VR-based education contents and technologies have already been introduced in medical education [3,4,5]. Currently, there is no doubt that online simulation trainings can provide nursing students with opportunities to experience knowledge-based cognitive thinking processes (e.g., problem-solving and knowledge integration), which thereby allow them to conduct clinical practice efficiently by overcoming the limitations of space related to physical training environments [29]; notwithstanding, a study showed that e-learning may actually halt improvement of communication skills by restricting direct relationship-building among participants [32], and another report demonstrated that e-learning methodologies should be reviewed owing to possibly evoking various long-term side effects [27]. The latter citation further demonstrated, by analyzing the existing VR-based nursing simulation training programs, that for VR-based e-learning to achieve a life-like virtual environment, there is the need to implement the active participation of clinical practitioners, and the need, among nursing students, for multidisciplinary cooperation and information technology for this methodology [27]. Furthermore, historically, and despite the rapidly evolving healthcare environment, the field of adult nursing has seen its simulation trainings remain largely limited to e-learning methodologies; thus, VR-based simulation programs, which increase students’ access to more realistic experiences during training, seem indispensable for this nursing field. Accordingly, the current study is significant because it assesses the needs of the future users of such VR-based simulation trainings—nursing students. We hope that the data we collected in this qualitative study provide relevant knowledge for stakeholders in the clinical competence of newly graduated nurses, and that our data help to foster the development of VR-based simulation training programs for adult nursing, a field that remains in an inchoate status.

In our results, students showed great expectations for VR-based simulation training. Specifically, they were hopeful about being able to practice various skills that are needed in clinical practice—without spatial and supply restrictions—in a more life-like environment. Furthermore, they wanted to experience a fun and engaging training through VR-based simulation, mainly by partaking, if possible, in care delivery for cases that are difficult to encounter in clinical practice, being able to access learning scenarios that have specific levels of difficulty, and receiving immediate feedback (e.g., like those in computerized games). In a recent study that administered an online VR-based simulation training for the care of acute cardiac diseases, the VR-based training group showed elevated nursing performance and learning confidence compared with the conventional simulation group; still, there were no differences in relevant knowledge, academic self-efficacy, and education satisfaction [29]. These results suggest that for VR-based simulation training programs in adult nursing to be more interesting for students, the developers of such programs should place greater focus on students’ needs, allow for a wider array of core skills to be practiced, and encompass scenarios that are more engaging.

In our study, participants proposed practical ideas that could be implemented in nursing curricula about the timing of VR-based simulation programs and specific methods by which the program could be implemented. Still, a study showed that such programs can have physical side effects (e.g., cybersickness), further demonstrating that such side effects were associated with program use duration, its visual design, and students’ movements while engaged in the program [33]. Hence, when developing VR-based simulation training programs, developers should, concomitantly, aim to reflect users’ needs, consider the effects of the visual components of the program, and make a thorough analysis of its educational elements; a prior study concurs with these statements [27].

To summarize based on our results, we deem that the development of an effective, realistic, and accessible VR-based simulation training program for adult nursing may require the active involvement of clinical specialists (i.e., to devise the scenarios), a clear assessment of users’ needs, and the collaboration between clinical specialists and IT experts (i.e., to ensure that the technical aspects are accurate). In this context, we believe that the present qualitative study contributed to this future development by providing knowledge on users’ needs (i.e., the needs of nursing students) regarding VR-based simulation programs for adult nursing. Future studies that study the opinions of clinical experts are warranted, as these may be conducive to the development of a VR-based simulation training program for adult nursing that meets both the institutions’ educational goals and the users’ needs.

## 5. Conclusions

In total, 40 themes emerged, divided into 13 theme clusters and the following four categories: “Limitations of clinical practice”, “Benefits of simulation training”, “Need to improve simulation training”, and “Need for VR-based simulation training.” Particularly, the themes that emerged in the theme cluster “Need for VR-based simulation training” were the following: “Great expectations for VR-based simulation training”, “Using desired scenarios during VR-based simulation training”, and “Various ways of running VR-based simulation training.” In other words, it was confirmed again that nursing students have limitations in clinical training and a need for improvement in simulation training. Recently, focusing on nursing educators, research on VR-based simulation training has begun to improve these limitations. From the results of this study, it was confirmed that nursing students also had high expectations and needs for VR-based simulation training. As shown by the results of the study, nursing students have high expectations for VR-based simulation training, have various opinions on VR simulation practice scenarios, and present various and realistic operating methods, confirming that students’ needs are large and specific.

Our results suggested that when developing an adult nursing VR-based simulation training program for nursing students, the development should aim to address the limitations of conventional clinical practice, and this can be done by clearly identifying such limits and students’ needs; a development focused on these gaps may allow for the optimization of the advantages of VR-based training and for specific and much needed improvements in the traditional simulation trainings to be dealt with. Furthermore, the program should be designed to reflect students’ various needs, which include the provision of the following: an array of scenarios and skills; difficulty-specific learning scenarios; immediate feedback (e.g., like those in computerized games); situations that simulate an emergency in nursing clinical practice; situations that simulate clinical cases that are difficult to experience in clinical practice; and a program that enables students to practice their patient–nurse communication skills.

To conclude, we propose that stakeholders should develop a VR-based simulation training program for adult nursing that reflects students’ needs.

## Figures and Tables

**Table 1 ijerph-17-08880-t001:** Questions for focus group interviews.

Categories	Questions
Introductory Questions	“Have you undergone clinical practice?” “Have you had simulation training in adult nursing?”
Transition Questions	“What were some of the challenges you had during clinical practice?” “How was your experience with simulation training?” “What were some limitations during simulation training?” “What did you deem as insufficient about simulation training?”
Key Questions	“What would you like to have experienced during clinical practice or simulation training?” “If a VR-based simulation training program was to be developed for adult nursing, what topics or situations would you deem as appropriate?” “Think about the aspects of clinical practice or simulation training that you felt needed improvement. If a VR simulation training program was to be developed, what aspects would require improvement?”
Ending Question	“Do you have any additional comments?”

**Table 2 ijerph-17-08880-t002:** Participants’ demographic characteristics (*n* = 14).

ID	Gender	Age	Total Clinical Experience (Number of Semesters/Subjects)	Adult Nursing Clinical Experience (Number of Subjects)	Simulation Training Experience (Number of Semesters)	Adult Simulation Training Experience (Number)
A	Female	23	4/6	4	2	5
B	Female	22	4/6	4	2	5
C	Male	26	4/6	4	4	7
D	Female	23	4/6	4	4	5
E	Female	22	4/6	4	4	5
F	Female	22	4/14	5	1	1
G	Male	26	4/12	4	2	4
H	Male	24	4/12	4	2	4
I	Female	22	4/14	5	1	1
J	Female	25	4/14	5	1	1
K	Male	25	4/14	5	1	5
L	Male	26	4/8	4	2	3
M	Female	22	4/14	5	1	5
N	Male	27	4/12	4	2	4

**Table 3 ijerph-17-08880-t003:** Needs for a VR-based simulation training program in adult nursing for nursing students.

Themes	Theme Clusters	Categories
▪ Feeling awkward for getting in the way of nurses’ work ▪ Patients do not want students’ participation in nursing care	▪ Feeling awkward	Limitations of clinical practice
▪ Flustered by an unexpected situation or care instruction ▪ Difficult to adapt to training because many parts were not covered at school	▪ Confounded	
▪ Limited training that only addresses some skills ▪ Only able to observe in most situations	▪ Limited training	
▪ Learning what is appropriate to do in clinical settings by hands-on practice ▪ Things learned in simulation training are helpful in clinical practice ▪ Learning in advance helped gain a different perspective during clinical practice	▪ Extremely helpful because it can be applied to clinical practice	Benefits of simulation training
▪ Able to give patients an explanation after simulation training ▪ Dealing with patients became easier after simulation training	▪ Become confident in clinical practice	
▪ Undeveloped nursing skills are discovered by hands-on practice ▪ Review mistakes through debriefing	▪ Improve on undeveloped nursing skills	
▪ Carelessly treat the unrealistic subjects in the simulation ▪ Subjects in the simulation cannot give feedback ▪ It is regrettable to be confined to only doing the actions determined according to the scenario	▪ Not so realistic	Need to improve simulation training
▪ Teaching method differs across instructors ▪ Competence varies across instructors	▪ Difference in effectiveness according to the instructor	
▪ Want to go to clinical practice only after learning how to deal with urgent situations ▪ Experiencing rare diseases helps during clinical practice ▪ Want to learn how to talk to patients and caregivers	▪ Want to experience diverse clinical situations	
▪ Not enough opportunities to undergo simulation training ▪ Want to be able to choose the desired case	▪ Want more simulation training opportunities	
▪ It will be helpful in learning skills ▪ Everything in the simulation feels real ▪ Diverse scenarios can be used, so it will be fun ▪ Relieve the gap in knowledge between theory and practice ▪ Standardized simulation training	▪ Great expectations for VR-based simulation training	Need for VR-based simulation training
▪ Partake in an environment just like the clinical setting ▪ Phased learning scenarios ▪ Immediate feedback, like in games ▪ Cases difficult to experience in clinical settings ▪ How to deal with urgent situations ▪ Practice core skills in real-world–like settings ▪ Practice communication ▪ Learn how to operate machines	▪ Using desired scenarios during VR-based simulation training	
▪ Want to learn how to deal with important situations in clinical practice prior to the actual training ▪ Want to undergo VR-based simulation training after experiencing the actual clinical setting ▪ Need to undergo simulation both before and after training ▪ Run simulation as an elective for advanced learning	▪ Various ways of running VR-based simulation training	

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
