# Peer review of "Needs Assessment for a VR-Based Adult Nursing Simulation Training Program for Korean Nursing Students: A Qualitative Study Using Focus Group Interviews"

_ijerph, 2020, doi:10.3390/ijerph17238880_

Round 1
Reviewer 1 Report
I enjoyed the opporuntunity to review your study!
As you’ll note below, I think it’s important to differentiate between “simulation training” and VR-based simulation training,” if they are indeed different. It was challenging to fully understand the rationale behind your FGI questions without having this in mind. For example, you asked participants “Think about the aspects of clinical practice or simulation training that you felt needed improvement. If a VR-simulation training program was to be developed, what aspects would require improvement?” This question suggests that the two are different, but an average reader may not know this.
I think there is an opportunity to be more explicit about your research question. I believe it is reflected in lines 89-91, but it is buried and the lengthiness of the presentation of categories, for me, made me have to remind myself of the explicit questions that this study was examining. To that point, while I think it is great that the authors are so transparent about their coding categories, the length of this section may call for some focused consideration of a smaller subset of these categories. In short, I'd recommend real attention be given to the presentation of the results -- there may be greater attention space given to how the conclusions drawn are supported by the results.
Comments on abstract:
Sentence “We used the FGI analysis method for data analysis” implies that there is a single method for analysis focus group data
In the sentence that reads “In total, 40 themes emerged, divided into ..” I might remove the category names from the abstract — the reader doesn’t have sufficient context at this point to make meaning of these category descriptors.
Lines 37-38: You write, “accordingly, these programs can be applied to a number of fields, depending on the contents.” It’s a bit unclear what the clause “depending on the contents” is suggesting.
Line 41: What does “contents” refer to and include?
Lines 73-74: You introduce the phrase “simulation training” — is this equivalent to training using VR? If so, you might want to indicate that the two are synonymous. If they are different, a sentence later in this paragraph or in the following that differentiates the two would be helpful.
Line 116 - You refer to Breen’s guidelines — if these are essential to your design, you may want to expand on this. Otherwise, it is necessary for readers who are unfamiliar to navigate to Breen’s article.
Line: 164-165: As above with the reference to Breen, it would be helpful to provide a brief explanation of “Colaizzi’s phenomenological methodology”
In table two, does “semester” refer to the number of semesters or denote the whether it was the individual’s first, second, third and so on semester
Reviewer 2 Report
The title and the introduction seem to be not represented by the structure and foci of the interview guide. What do I mean - the title and introduction are very focused on the use of VR, while the interview guide seems more focused on general simulation. A needs assessment is focused on exactly that, do we need something, rather this article is focused on how to improve the VR simulation. I am not certain there is a total connect to the desrired goals of this research to the design and methods that were presented.
This paper is noted to be about VR, but yet the overall questioning
that was done was based around general clinical training. Also, the quest
ions that did pertain to VR were more focused on a wish list rather than
a needs assessment. But this needs assessment was applied to Nursing
students who not necessarily even knew what VR was, which makes this a
very tough study to garnish any good results. The struggles that were
listed in the introduction are struggles across Nursing programs, but
it doesn't necessarily lead to the VR interventions. This is what
makes this study flawed. What about general simulation, this is a
cheaper and easier form of education that has been validated for years?
General simulation may be more appropriate for this population of
nursing students. But yet the authors in the went straight to the big
guns, VR simulation. Without any prior exposure to VR or VR
simulations, how would the students be able to know if they need VR or
VR was the best possible option. To me it just seems disjointed and
the questions in the interview guide seemed to be very specific,
rather to gather the required information for a needs assessment.
Rather this seemed to be a wish list, rather than a needs assessment.
Round 2
Reviewer 2 Report
With the minor additions the manuscript has been improved.